# Genome-Wide Association Study Reveals Genetic Architecture and Candidate Genes for Yield and Related Traits under Terminal Drought, Combined Heat and Drought in Tropical Maize Germplasm

**DOI:** 10.3390/genes13020349

**Published:** 2022-02-15

**Authors:** Alimatu Sadia Osuman, Baffour Badu-Apraku, Benjamin Karikari, Beatrice Elohor Ifie, Pangirayi Tongoona, Eric Yirenkyi Danquah

**Affiliations:** 1West Africa Centre for Crop Improvement (WACCI), University of Ghana, PMB 30 Legon, Accra 00223, Ghana; asosuman@wacci.ug.edu.gh (A.S.O.); bifie@wacci.ug.edu.gh (B.E.I.); ptongoona@wacci.ug.edu.gh (P.T.); edanquah@wacci.ug.edu.gh (E.Y.D.); 2International Institute of Tropical Agriculture (IITA), PMB 5320, Ibadan 200001, Nigeria; 3Crops Research Institute, P.O. Box 3785, Kumasi 00223, Ghana; 4Department of Crop Science, Faculty of Agriculture, Food and Consumer Sciences, University for Development Studies, P.O. Box TL 1882, Tamale 00223, Ghana; benkarikari1@gmail.com

**Keywords:** candidate genes, climate smart, genomics, marker-assisted selection, sub-Saharan Africa

## Abstract

Maize (*Zea mays* L.) production is constrained by drought and heat stresses. The combination of these two stresses is likely to be more detrimental. To breed for maize cultivars tolerant of these stresses, 162 tropical maize inbred lines were evaluated under combined heat and drought (CHD) and terminal drought (TD) conditions. The mixed linear model was employed for the genome-wide association study using 7834 SNP markers and several phenotypic data including, days to 50% anthesis (AD) and silking (SD), husk cover (HUSKC), and grain yield (GY). In total, 66, 27, and 24 SNPs were associated with the traits evaluated under CHD, TD, and their combined effects, respectively. Of these, four single nucleotide polymorphism (SNP) markers (SNP_161703060 on Chr01, SNP_196800695 on Chr02, SNP_195454836 on Chr05, and SNP_51772182 on Chr07) had pleiotropic effects on both AD and SD under CHD conditions. Four SNPs (SNP_138825271 (Chr03), SNP_244895453 (Chr04), SNP_168561609 (Chr05), and SNP_62970998 (Chr06)) were associated with AD, SD, and HUSKC under TD. Twelve candidate genes containing phytohormone cis-acting regulating elements were implicated in the regulation of plant responses to multiple stress conditions including heat and drought. The SNPs and candidate genes identified in the study will provide invaluable information for breeding climate smart maize varieties under tropical conditions following validation of the SNP markers.

## 1. Introduction

The world’s population is estimated to reach 10 billion by 2050 [1]. This, together with reduction in arable land and climate change will lead to a rise in abiotic and biotic stresses which are the major threats to food and nutritional security. Maize is an important food security crop and is crucial for economic development in many developing countries in Asia, Latin America, and sub-Saharan Africa [2]. The savannas of sub-Saharan Africa (SSA) contribute to high maize productivity through high incoming solar radiation, reduced incidence of pests and diseases due to low humidity, and low night temperature [3,4]. Additionally, climate change threatens the goal of achieving global food security and could have severe socio-economic consequences globally [5]. With the fast increasing world population, maize production and productivity are expected to be significantly reduced by the adverse impacts of climate change and lead to a global food crisis which is expected to have major impacts in SSA in particular [6]. The effects of adverse climatic conditions such as high temperatures, erratic rainfall patterns, and drought could have serious impacts on maize production and productivity and thus significantly reduce global food production [7,8]. Although maize is well adapted and substantially produced and utilized in the savannas of SSA, the simultaneous incidence of abiotic stresses such as drought and high temperature during the maize flowering period could reduce the photosynthetic rate, accelerate leaf senescence, induce kernel abortion, and ultimately result in drastic yield losses [9,10]. The combination of the two stresses could lead to a grain yield loss of more than 90% during the flowering and grain filling period in maize [4,11,12,13,14]. Thus, the available evidence from climate projections indicates decreasing precipitation, increasing temperatures, and high intensity and frequency of heat and drought stresses [15,16,17]. Nelimor et al. [18] evaluated 33 landraces from Burkina Faso (6), Ghana (6), and Togo (21), and three drought-tolerant populations/varieties from the Maize Improvement Program at the International Institute of Tropical Agriculture (IITA) under three conditions, namely managed drought stress, heat stress, and combined drought and heat stress, with optimal growing conditions as control, for two years. The results indicated that phenotypic and genetic correlations between grain yield of the different treatments were very weak, suggesting the presence of independent genetic control of yield to these stresses. However, grain yield under heat and combined drought and heat stresses were highly and positively correlated, indicating that heat-tolerant genotypes would most likely tolerate combined drought and heat stress. Yield reduction averaged 46% under managed drought stress, 55% under heat stress, and 66% under combined drought and heat stress, which reflected the hypo-additive effect of drought and heat stress on grain yield of the maize accessions.

Heat stress is reported to prolong the anthesis–silking interval, reduce kernel set [9,10,19,20], decrease photosynthetic rate and chlorophyll content [14,19,21], and lead to damaged cellular membrane [22]. Conversely, drought and or terminal drought at seedling stage has deleterious impacts on seedling establishment, vegetative growth, photosynthesis, root growth, anthesis, anthesis–silking interval, pollination, and grain formation in maize [23,24]. Combined heat and drought is well documented to reduce photosynthetic efficiency, stomatal conductance, leaf area, water use efficiency, and ultimately grain reduction [25,26]. There is more propelling evidence that the combined effect of heat and drought on plant growth and productivity is more severe than the individual effects of these stresses in several crops including maize [25,27,28,29,30,31]. Briefly, combined heat and drought is achieved when the field/plants receive adequate water from planting up to 2–3 weeks before anthesis [32,33], while terminal drought occurs when the field/plants receive adequate water every week from planting up to 32 days after planting [33,34]

The majority of traits which are of agronomic importance in maize are quantitatively inherited, thus, they are controlled by multiple genes/loci with both major and minor effects, and are largely influenced by the environment [35,36]. Two strategies are available for dissecting such traits genetically: biparental/linkage mapping and genome-wide association study (GWAS) [37,38]. The GWAS strategy has become more popular in recent years due its comparative advantages over the biparental or linkage mapping method [39,40]. The advantages of the two methods include higher mapping resolution, more allelic diversity with less time required for population development [39,41,42]. Additionally, recent reduction of sequencing cost by the next-generation sequencing (NGS) technologies has made application of GWAS in mapping of quantitative trait loci (QTLs) more popular in identifying markers/single nucleotide polymorphisms (SNPs) that cause variation in the phenotype [40].

Several QTLs have been identified via GWAS in maize including QTLs associated with drought or heat tolerance [12,13,14,43,44,45,46,47]. However, reports of QTLs linked to both heat tolerance and combined heat and drought tolerance are limited [11]. Longmei et al. [14] detected a total of 30 SNPs strongly associated with grain yield (GY), anthesis–silking interval (ASI), and days to 50% anthesis (AD) under heat stress conditions. Yuan et al. [11] detected 549 SNPs that were significantly associated with 12 traits under one or combined environments (well-watered, drought and heat stress conditions). Of these SNPs, S1_140960558 on chromosome 1 was associated with days to anthesis under the three conditions while 76 SNPs were linked to grain yield under both drought and heat stresses. These suggested the complex nature of the traits. Drought and heat stresses are known to affect growth and yield of maize significantly as well as predispose the grains to pre-harvest aflatoxin contamination [48,49]. Therefore, understanding the genetic architecture of a trait of interest significantly influences the breeding strategy that will be most effective [50] for genetic enhancement. 

Breeding for heat stress as well as drought tolerance in maize is crucial to the effort to overcome the challenges associated with these two key threats to sustainable maize production. Therefore, a total of 162 tropical maize inbred lines developed by the IITA were evaluated under two contrasting environments: heat combined with drought (CHD), and terminal drought (TD) conditions. Furthermore, the inbred lines were genotyped using the genotyping-by-sequencing (GBS) with the Diversity Array Technology (DArT) sequencing (DArTseq) [27,28,29]. The objectives of the study were to (i) map the genomic regions associated with GY and its related traits under CHD conditions, (ii) identify markers linked to the various traits under TD conditions, and (iii) predict the putative candidate genes underlying the genomic regions of the studied traits under stress conditions. The findings from this study will be useful for marker-assisted breeding for the development of resilient maize varieties. Additionally, the predicted putative candidate genes could form the foundation for future functional validations to unravel the molecular mechanisms involved in the response of maize to CHD and TD. 

## 2. Materials and Methods

### 2.1. Genetic Resources, Field Evaluation, and Phenotyping 

In total, 162 early maturing inbred lines obtained from IITA, Ibadan, Nigeria were used for this study. The lines were evaluated at the Manga Station of the Savannah Agricultural Research Institute, Ghana (11°00.977 N, 000°15.912 W) and Kadawa, Nigeria (11°450 N, 8°450 E) in 2018 and 2019 under CHD and TD conditions (Figure 1). Detailed descriptions of the panel of inbred lines used, treatments (CHD and TD), and traits measured have been published in our previous study [33]. Briefly, the maize plants were exposed to DH. Data were recorded on the following: days to 50% anthesis (AD), days to 50% silking (SD) as the number of days from planting to when 50% of the plant had extruded pollen and produced silks, respectively; anthesis–silking interval (ASI) was recorded as the difference between SD and AD; stalk lodging (SL) was recorded as percentage plants leaning more than 30° from vertical; husk cover (HUSKC) was recorded 3 weeks post-flowering using a scale of 1–9 (see Appendix A); plant aspect (PLASP) was measured based on the overall assessment of the architecture of the plant in a plot as they appeal to sight using a scale of 1–9 (see Appendix A); leaf death (LD) was determined by estimating the percentage of leaves dead from the base of the plant to the flag leaf using a scale of 1–9 (see Appendix A) at 70 days after planting (DAP); plant height (PLTH) and ear height (EARH) were measured on 10 randomly selected plant from each plot, as a distance from the bottom of the plant to the first tassel branch and to the height of node bearing the upper ear, respectively; root lodging (RL) was measured as the percentage plant bent below or at the highest ear node; ear rot (EARO) was measured by counting the number of ear showing signs of rot within a plot; ear aspect (EASP) and ears per plant (EPP) were taken at harvest. EASP was measured using the general appearance of the ear without the husk using a scale of 1–9 based on the ear size (see Appendix A), uniformity of the size, texture, extent of grain filling, insect, and disease damage. Tassel blasting (TB) was measured as the total number of plants with tassel blast and leaf firing (LF) was measured as the total number of plants with leaf firing at the upper part of the plant, 1–2 weeks after silking and tasseling. Grain yield (GY kg/ha) was measured from the shelled grain weight per plot and adjusted to 15% moisture content as outlined by Longmei et al. [14] and Nelimor et al. [18]. The temperature and rainfall data for the experiments at the two test locations were recorded.

### 2.2. Statistical Analysis 

Descriptive statistics: mean, standard deviation (SD), range (minimum–maximum), and coefficient of variation (CV%) of the phenotypic data were computed using SAS version 9.3 (SAS Institute, 2010, Inc., Cary, NC, USA). The frequency distribution and correlation analyses were performed using Performance Analytics R package [51]. Combined analysis of variance was conducted for each of the traits under CHD and TD conditions using a generalized linear model as outlined by Karikari et al. [37]. The broad-sense heritability (*H*^2^) was calculated following the formula by [52], $H^{2}=\sigma_{g}^{2}/\left(\sigma_{g}^{2}+\frac{\sigma_{ge}^{2}}{n}+\frac{\sigma_{e}^{2}}{nr}\right)$, where $\sigma_{g}^{2}$ is the genotypic variance, $\sigma_{ge}^{2}$ is the genotype by environment interaction variance, $\sigma_{e}^{2}$ is the error variance, e is the number of environments, and r is the number of replications.

### 2.3. Genome-Wide Association Study Mapping

The total DArT SNP markers for this panel was 9684 as earlier reported in our study [33]; further filtering was undertaken in pLINK v1.07 with the indep-pairwise 50 10 0.5 command option [53], leaving 7834 SNPs of higher quality.

Association analysis among the SNPs and traits was performed using the mixed linear model (MLM) implemented in Traits Analysis by Association, Evolution and Linkage (TASSEL) version 3.2.3.1 (Figure 1) [54]. The MLM adopted was proposed by Yu et al. [55] with each molecular marker considered a fixed effect and assessed individually: $$Y=\;X_{\beta }+W_{\alpha }+Q_{V}+z_{U}+\varepsilon$$
where Y is the observed vector of means; *β* is the fixed effect vector (p × 1) other than molecular marker effects and population structure; α is the fixed effect vector of the molecular markers; *ν* is the fixed effect vector from the population structure; u is the random effect vector from the polygenic background effect; *X*, *W*, and *z* are the incidence matrices from the associated *β*, *α*, *ν*, and *u* parameters; and ε is the residual effect vector. This model takes into consideration population structure (Q) and relatedness (K), hence, the Q matrix obtained in the population structure analysis was the same as the one reported earlier by Osuman et al. [33] while the K matrix computed in TASSEL was used. Additionally, to reduce false positive associations, the Bonferroni correction −log*P* (1/m), where m = number of SNPs, was adopted as the threshold, −log*P* (1/7834) ≈ 3.89 and this was equivalent to *p*-value = 1.27 × 10^−4^ for declaring the significance of marker–trait association (MTA). Some selected trait association results were visualized using the Manhattan and quantile–quantile (QQ) plots in R with package *CMplot* (https://github.com/YinLiLin/R-CMplot, accessed on 25 October 2020).

### 2.4. Candidate Gene Prediction and In Silico Analyses

Model genes within ±120 kb of stable SNP positions (SNPs detected under either CHD and TD or across (combined) effects (Figure 1) were downloaded from B73 reference genome (version 4) via the online platform qTeller MaizeMGB (https://qteller.maizegdb.org/, accessed on 14 December 2021) [56]. The expression of model genes was retrieved from qTeller MaizeMGB with control and drought leaf deposited by Forestan et al. [57] and, control and heat treated seedlings by Waters et al. [58], and the expression profiles retrieved were heatmapped via TBtool [59]. The functional annotations such as gene ontology (GO) [60], protein families (Pfam) [61], and Kyoto Encyclopedia of Genes and Genomes (KEGG) [62] of the model genes ±120 kb of stable SNP positions were retrieved from Phytozome version 13 available on https://phytozome-next.jgi.doe.gov/, accessed on 14 December 2021) [63]. Promoter regions of the model genes were analyzed for Cis-acting regulatory elements (CAREs) that may be involved in modulating plant response under CHD and TD conditions with 1.5 kb upstream of the start codon (ATG) in PlantCare database [64]. Based on the expression profiles, annotations, and CAREs retrieved, potential candidate genes were predicted. Gene structure was visualized via the online platform Gene Structure Database Server (http://gsds.gao-lab.org/, accessed on 14 December 2021).

## 3. Results

### 3.1. Phenotypic Variation, Correlation, and Heritability among the Tropical Maize Germplasm 

The rainfall and temperature recorded during the experiments are presented in Appendix A. The frequency distribution, scatter plot, and correlation among the quantitative traits evaluated/computed are shown in Appendix A under CHD and TD, respectively. All the agronomic traits measured under the test environments showed continuous and normal distributions which is typical of quantitative traits. For example, the average of the two years phenotypic data from the two countries under CHD experiments had: AD, SD, and GY mean ± standard error (range) values of 66.85 ± 0.26 days (48.16–75.73 days), 67.66 ± 0.27 days (49.28–77.73 days), and 623.60 ± 23.95 kg/ha (94.24–1684 kg/ha), respectively (Appendix A). Under terminal drought condition on the other hand, mean ± standard error (range) values for AD, SD, and GY of 48.09 ± 0.35 days (23.13–53.22 days), 52.97 ± 0.40 days (25.09–59.91 days), and 1406.00 ± 41.70 kg/ha (310.40–6464 kg/ha), respectively (Appendix A) were recorded. These results suggest that CHD and TD stresses alter the phenological, growth, and yield related traits in maize.

Positive and significant correlations (*p* < 0.05, *p* < 0.01, *p* < 0.001) were observed between AD and SD (r = 0.97), HUSKC and PLASP (r = 0.59), and PLASP and EASP (r = 0.54) under CHD (Appendix A). Similarly, under TD conditions, AD and SD, AD and HUSKC, SD and HUSKC, and EASP and EAROT had a strong correlation (r < 0.60) (Appendix A). These revealed that these traits as well as others with positive correlations could be selected without any negative effect on the other traits. The majority of the measured traits differed significantly among the lines (genotypes), environment, and genotype by environment interaction (GxE) (Appendix A). Average $H^{2}$ of the traits was ≈45.54% with the minimum and maximum values of 4.70% (in Manga, Ghana) and 82.00% (in Kadawa, Nigeria), respectively, both under CHD conditions (Appendix A). These results indicated that the measured traits may have been influenced by either genetic or environmental factors or both. 

### 3.2. Marker–Trait Association under Combined Heat and Drought Conditions

A total of 15 traits (AD, ASI, EARO, EASP, EPP, HUSKC, LD, LF, PLASP, PLTH, RL, SD, SL, TB, and GY) were used with the 7834 SNPs taking into consideration the population structure and relatedness earlier published [33]. Sixty-six SNPs were significantly associated with the 15 traits at the threshold ≥3.89 (Figure 2a–d; Appendix A). These were distributed across the 10 chromosomes in maize genome with −log*P* and phenotypic variation explained (R^2^) ranging from 3.89 to 15.55% and 8.37–49.23%, respectively. Among these, the highest number of 13 and 10 SNPs was found on chromosomes (Chr) 1 and 4, respectively. This indicates the major role of Chr01 in regulating the traits evaluated under CHD conditions in maize (Figure 2a). In all, 15 SNP marker–trait associations (MTA) signaling for GY were detected on Chr01 (SNP_81895008 and SNP_82684770), Chr02 (SNP_42619227), Chr03 (SNP_6825039 and SNP_143903954), Chr04 (SNP_42991074, SNP_49826316, SNP_168609387, SNP_213750918, and SNP_231684692), Chr05 (SNP_193737891 and SNP_221804675), Chr09 (SNP_22059270), and Chr10 (SNP_103783517 and SNP_126092807). For details on their respective −log*P* and R^2^ see Appendix A. These indicate that Chr04 is the hotspot for GY (Figure 2a). Additionally, two SNPs (SNP_84602389 and SNP_166466778) on Chr04 were associated with EPP (Appendix A). Interestingly, four repetitive SNPs, SNP_161703060 on Chr01, SNP_196800695 on Chr02, SNP_195454836 on Chr05, and SNP_51772182 on Chr07, were detected concurrently to be associated with the flowering traits AD and SD (Appendix A). 

### 3.3. Marker–Trait Associations under Terminal Drought Condition

Ten traits viz., AD, EARH, EARO, EASP, EPP, HUSKC, RL, SD, SL, and GY were used for mapping for MTAs employing Q + K in TASSEL. A total of 27 SNPs were linked to these traits with −log*P* and R^2^ ranges of 3.90–24.11% and 12.53–64.79%, respectively, across the chromosomes except Chr10 (Figure 2b; Table 1). The highest number of SNPs was detected on Chr02 (6 SNPs), Chr05 (5 SNPs), and Chr06 (5 SNPs) followed by Chr03, with 4 SNPs, Chr01 and Chr04 with 2 SNPs each while Chr07, Chr08, and Chr09 recorded 1 SNP each (Figure 2b; Table 1). This showed that Chr02, Chr05, and Chr06 are hotspots for modulating maize responses under terminal drought conditions. However, the conspicuous peak of the SNP (SNP_269120178) on Chr01 with −log*P* (24.11) and R^2^ (64.11%) together with the other two SNPs (SNP_104170418 and SNP_162101608) on Chr02 were linked to GY (Figure 2b; Table 1). In addition, 2 SNPs, SNP_51109816 on Chr02 and SNP_99059711 on Chr03, were linked to EPP (Table 1). Two markers, one on Chr02 (SNP_3121382) and the other on Chr04 (SNP_199184898), had pleiotropic effects on AD and SD. Moreover, the four SNPs, namely SNP_138825271 (Chr03), SNP_244895453 (Chr04), SNP_168561609 (Chr05), and SNP_62970998 (Chr06), were associated with AD, SD, and HUSKC concurrently, with each SNP accounting for an average of 17.27% of the variations in the three traits (Table 1).

### 3.4. Mapping Using Data Averaged across Combined Heat and Drought Conditions plus the Terminal Drought Conditions

Eleven common traits selected across CHD and TD conditions, i.e., AD, ASI, EARO, EASP, EPP, LD, PLASP, RL, SD, SL, and GY, with means across these conditions (CHD and TD) were used for MTA mapping. A total of 24 SNPs with a mean number of ≈2.8 SNPs, minimum = 1 SNP (on either Chr07 or Chr08), and maximum = 5 SNPs (on Chr04) were associated with the 11 traits (Figure 2c; Table 2). These SNPs peaked in the range of 3.94–15.57% and caused 13.05–47.77% phenotypic variation (Table 2). Three SNPs (SNP_203396231, SNP_265278865, and SNP_269120178) on Chr01 were linked to GY, whereas two SNPs (SNP_112491022 and SNP_99059711) were associated with EPP (Figure 3 and Figure 4; Table 2). The hotspot for EASP was on Chr04 with 4 SNPs followed by 3 and 2 SNPs for GY on Chr02 and Chr01, respectively, and 2 SNPs on Chr02 associated with ASI (Figure 2c; Table 2). One repetitive SNP (SNP_11473743) on Chr01 associated with both AD and SD (Table 2).

### 3.5. Comparison of Mapping Results from the Three Conditions (CHD, TD, and Combined)

Cumulatively, 117 SNPs were detected among the traits under the three environmental conditions, CHD, TD, and across (combined) mapping (Figure 4). No SNP was detected under the combined three conditions (Figure 4). However, 3 SNPs (SNP_11473743, SNP_51772182 and SNP_11473743) were detected under CHD and across (combined) conditions (Figure 5, Table 3). The three SNPs: SNP_11473743 (Chr02) was linked to both AD and SD across conditions (combined) and AD under CHD condition, SNP_51772182 (Chr07) was linked to PLASP across conditions (combined) and also linked to EASP under CHD conditions, and SNP 11473743 (Chr09) was lined to LD (Table 3) Nine SNPs (SNP_269120178, SNP_27502516, SNP_24521844,SNP_99059711, SNP_82875264, SNP_164264714, SNP_125871560, SNP_43475091, and SNP_89731392) were detected under TD conditions as well as across conditions (combined) (Figure 5; Table 3). Four of these SNPs, SNP_269120178 (Chr01), SNP_99059711 (Chr03), SNP_125871560 (Chr06), and SNP_43475091 (Chr08) were linked to GY, EPP, RL, and SL, respectively. Additionally, two SNPs (SNP_27502516 and SNP_24521844) on Chr02 were associated with EARO while two other SNPs (SNP_82875264 and SNP_164264714) on Chr05 were associated with RL.

### 3.6. Candidate Gene Prediction and In Silico Analysis 

With availability of reference genomes and user-friendly bioinformatics tools for in-silico prediction, further downstream analyses have become feasible in recent years [65]. Candidate genes are of relevance for functional validation to unearth the regulatory mechanisms underlying each trait of interest. Therefore, we used the 12 common SNPs to retrieve models within the interval position of the SNPs (±120 kb) (Table 3) via reference genome (B73 v4) through online platform qTeller MaizeMGB (https://qteller.maizegdb.org/, accessed on 15 December 2021) [56]. In all, we identified 55 model genes within 12 SNP positions ±120 kb with varied expression under drought and optimal growing conditions [57] or heat and control conditions [58] (Appendix A); out of these, 12 potential candidate genes were predicted (Table 4). With the exception of *Zm00001d041124* located at 99054551–99065304 bp on Chr03, the remaining 11 genes have annotations related to either CHD/TD or both conditions (Table 4). For instance, *Zm00001d048531* located 117.36 kb (downstream) from SNP_158056460 encodes RNA helicase which has been implicated in improving abiotic stress tolerance in crop plants [66,67]. Most of the predicted candidate genes contain CAREs essential to regulate gene expression under CHD and TD conditions (CGTCA/TGACG (MeJA-responsiveness); ABRE (Abscisic acid); circadian, P-box (Gibberellin responsiveness); AuxRR-core/TGA (auxin); TCA (Salicylic acid); TC-rich repeats (defense and stress responsiveness); (Appendix A). Candidate gene structure analyses revealed that *Zm0001d002374* possessed 1 exon with no intron while the *Zm00001d046434* possessed 17 exons and 16 introns (Figure 6).

## 4. Discussion

During the last century, conventional breeding approaches have contributed to the development of several high-yielding and superior quality maize breeds [68,69,70,71]. However, the progress achieved so far via conventional approaches is not enough to feed the growing population in the midst of climate change, reduction in arable land, and declining soil fertility. Breeding via conventional approaches is time consuming, expensive, and also phenotypic selection alone does not always result in the development of donor materials with the best potential for trait diversification and improvement. Hence, the present study was conducted to identify SNPs that are linked to the numerous agronomic traits under either CHD or TD conditions or both conditions for marker-assisted breeding. 

Under CHD conditions, a total 66 SNPs associated with the 15 measured traits, i.e., AD, ASI, EARO, EASP, EPP, HUSKC, LD, LF, PLASP, PLTH, RL, SL, SD, TB, and YIELD (at Bonferroni-correction −log10(*P*) ≥ 3.89) were found across the 10 chromosomes in the maize genome (Figure 2a–d; Appendix A). Either of these SNPs caused 8.37–49.23% variation in phenotype. This indicated that the 15 traits were complex in nature with numerous loci affecting the phenotypes. The hotspot for YIELD in this study was on Chr04 (SNP_42991074, SNP_49826316, SNP_168609387, SNP_213750918, and SNP_231684692). This is not surprising as Yuan et al. [23] previously reported 4 SNPs for grain yield under heat stress condition. Similarly, chromosomes 2 and 7 were hotpot regions for SL and Chr06 for RL (Figure 2a–d; Appendix A). These pinpoint the important roles of these chromosomes in regulating maize response under CHD conditions. A number of earlier studies have reported that several traits such as AD and ASI usually exhibit strong genetic correlations with grain yield [43,72,73] but in this study weak and positive significant correlations (r = 0.25) were observed (Appendix A). In this study, four SNPs, SNP_161703060, SNP_196800695, SNP_195454836, and SNP_51772182 on chromosomes 1, 2, 5, and 7, respectively, showed pleiotropic effects on both AD and SD (Appendix A). This implied that the four SNPs regulated the two important traits simultaneously and could be targeted for MAS to improve these traits concurrently. This could be the basis for the strong correlation (r = 0.97) coefficient between AD and SD. Flowering in maize (denoted as AD and SD in this study) is an important trait in breeding for drought/heat tolerance. In this study, these two traits were concurrently identified to be linked together, emphasizing their importance in selecting for maize tolerant to drought and combined heat and drought. For example, early flowering days resulted in shorter anthesis–silking interval, increased plant and ear height, increased EPP, and delayed senescence [74]. Additionally, when maize flowers under drought conditions, there is a delay in silking and the interval between anthesis and silking increases thereby giving rise to a longer anthesis–silking interval (ASI) [75]. 

Terminal drought stress is one of the most vital environmental stress factors that cause a significant reduction in maize productivity. The present study identified markers linked to 10 traits of agronomic significance viz, AD, EARH, EARO, EASP, EPP, HUSKC, RL, SD, SL, and YIELD. Twenty-seven SNPs were associated with these 10 traits at the peak of 3.90–24.11 with 12.53–64.79% phenotypic variation in all chromosomes except chromosome 10 (Table 1). This implies that breeding by traditional conventional approaches may result in lower genetic gain since these traits are regulated by both major and minor loci [76]. The most prominent SNP (SNP_269120178) on chromosome 1 with −log*P* (24.11) and R^2^ (64.11%) plus two other SNPs (SNP_104170418 and SNP_162101608) on chromosome 2 were linked to YIELD (Table 1. Two SNPs (SNPs: SNP_51109816 and SNP_99059711) were linked to EPP (Table 1). These five SNPs could be targeted for MAS to improve YIELD and ear aspect-related traits under terminal drought condition since grain yield is the key output of production in agriculture. To the best of our knowledge, no GWAS has been reported for grain yield and other secondary traits in terminal drought. Therefore, the information provided in the present study could be useful for breeding for tolerance to terminal drought.

Stability of QTLs/SNPs across multiple environments and genetic backgrounds is a requisite for their utilization in practical plant breeding [77,78]. Hence, the means of the traits (AD, ASI, EARO, EASP, EPP, LD, PLASP, RL, SD, SL, and YIELD) under CHD and TD were used for the mapping of the genes. In all, 24 SNPs associated with these traits across nine chromosomes were used in the present study. Three SNPs on Chr01 (SNP_203396231, SNP_265278865, and SNP_269120178) and 2 SNPs on Chr03 (SNP_112491022 and SNP_99059711) were linked to YIELD and EASP, respectively (Figure 3 and Figure 4; Table 2). These five SNPs could be targeted for allele pyramiding aimed at improving these traits simultaneously. Additionally, the results of the mapping with the means across CHD and TD conditions revealed that some SNPs were detected across either CHD or TD and combined conditions (means) (Table 3). These SNPs are considered as stable and are recommended for further validation and use in practical maize breeding programs. In all, 105 SNPs were associated with the traits evaluated under the three scenarios. These SNPs could be useful in future genomic selection breeding programs targeted at all the traits at once. 

In an effort to select potential candidate genes for cloning and functional verification using gene overexpression and CRISPR/Cas technology, a number of candidate genes were predicted based on propelling evidence from in silico and could be validated to unravel their actual regulatory role in the studied traits. For instance, recent evidence suggest that Auxin induced-like protein (*Zm00001d002374*) predicted in the present study is reported to play a significant role under drought or heat and combined heat and drought stress in Arabidopsis [79,80]. Circadian clock and CARES are known to regulate gene expression under different conditions [81]. It is of great interest to identify the candidate genes underlying the genomic region for practical plant breeding to unravel the molecular mechanism underlying any trait of interest [38], therefore, the 12 candidate genes predicted together using the CAREs, expression profiles, and gene structure analyses (Table 4 and Appendix A; Figure 6) provided valuable insight for future functional verification experiments/projects. 

## 5. Conclusions

Multiple traits from 162 inbred lines and 7834 high quality SNPs were used to conduct MTAs. In all, 117 SNPs associated with the measured traits were evaluated in the present study. Specifically, 66, 27, and 24 SNPs associated with the traits were evaluated under CHD, TD, and combined CHD and TD conditions, respectively. Of these, three SNPs were repetitive under CHD and combined CHD and TD conditions and nine were found concurrently between TD and combined conditions. The highest number of SNPs detected under CHD conditions elucidated the complex nature of the genetic factors that regulated the maize responses under the test conditions. The stable SNPs and those linked to more than one trait could be targeted for MAS. Twelve candidate genes as well as their CAREs, gene structure, and expression profiles were predicted by bioinformatics analyses. The findings from this study offer useful clues about the genetic architecture of these traits under multiple abiotic stresses. The results of this study provided essential information that could be used to improve maize breeding programs in SSA.

## Figures and Tables

**Figure 1 genes-13-00349-f001:**
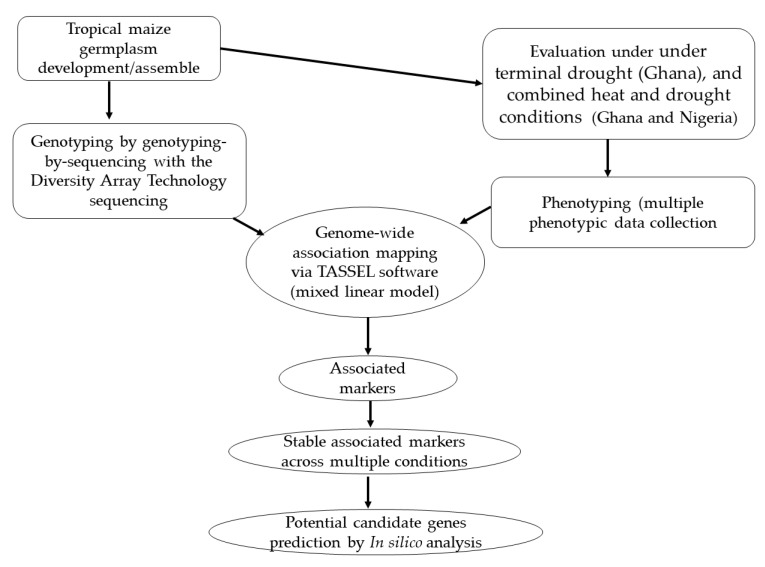
Flowchart of the study.

**Figure 2 genes-13-00349-f002:**
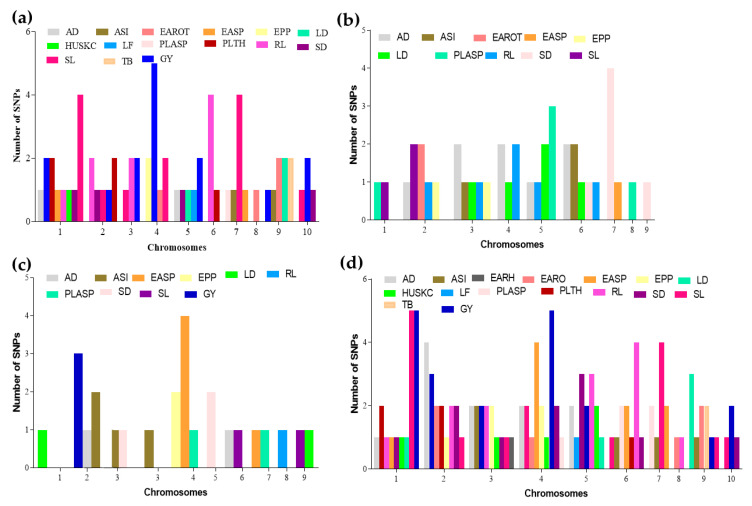
Number of significant SNPs detected per chromosome in (**a**) combined heat and drought conditions (CHD); (**b**) terminal drought conditions (TD); (**c**) average of CHD and TD conditions. (**d**) Total number of significant SNPs detected per chromosome in the study. Days to 50% anthesis (AD), days to 50% silking (SD), anthesis silking interval (ASI), stalk lodging (SL), husk cover (HUSKC), plant aspect (PLASP), leaf death (LD), plant height (PLTH), root lodging (RL), ear height (EARH), ear rot (EARO), ear aspect (EASP), ears per plant (EPP), tassel blasting (TB), leaf firing (LF), and grain yield (GY).

**Figure 3 genes-13-00349-f003:**
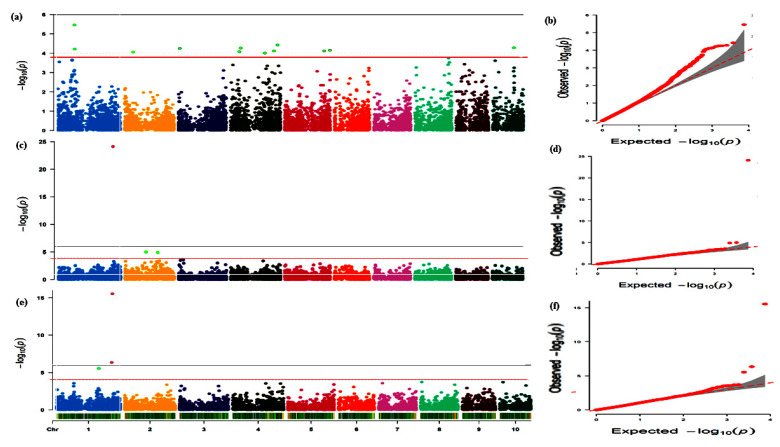
Manhattan plots (left) and quantile–quantile (QQ) plots (right) for grain yield across the three treatments. (**a**,**b**). Manhattan and QQ plot under combined heat and drought conditions (CHD) (**c**,**d**). Manhattan and QQ plot under terminal drought conditions (TD) (**e**,**f**). Manhattan and QQ plot for means obtained from CHD and TD. The threshold of 3.89 (Bonferroni correction) was adopted with the red line in the Manhattan plots. The X-axis represents chromosome number while the Y-axis represents $-\log_{10}\left(P\right)$. The X and Y axes of the QQ plots represent the expected and observed $-\log_{10}\left(P\right)$, respectively. The red line in the QQ-plots with the shaded regions indicates a 95% confidence interval.

**Figure 4 genes-13-00349-f004:**
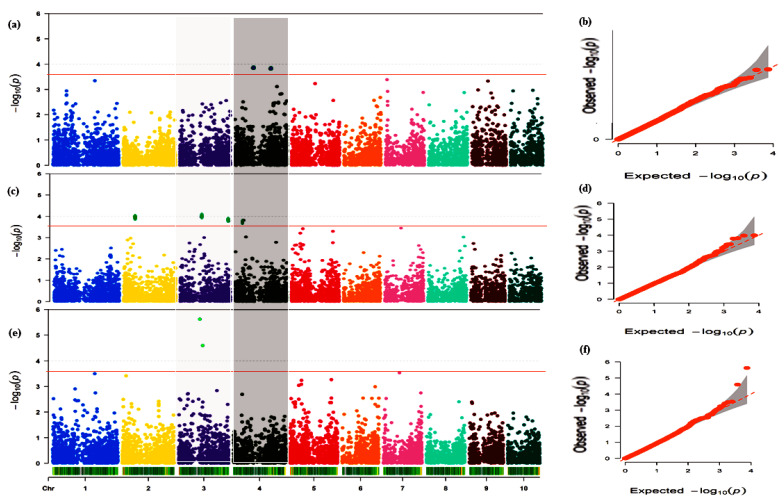
Manhattan plots (left) and quantile–quantile (QQ) plots (right) for ears per plant across the three treatments (**a**,**b**). Manhattan and QQ plot under combined heat and drought conditions (CHD). (**c**,**d**). Manhattan and QQ plot under terminal drought conditions (TD) (**e**,**f**). Manhattan and QQ plot for means obtained under CHD and TD. The threshold of 3.89 (Bonferroni correction) was adopted with the red line in the Manhattan plots. The X-axis represents chromosome number while the Y-axis represents $-log_{10}\left(P\right)$. The X and Y axes in the QQ plots represent the expected and observed $-\log_{10}\left(P\right)$, respectively. The red line in the QQ-plots with the shaded regions indicates a 95% confidence interval.

**Figure 5 genes-13-00349-f005:**
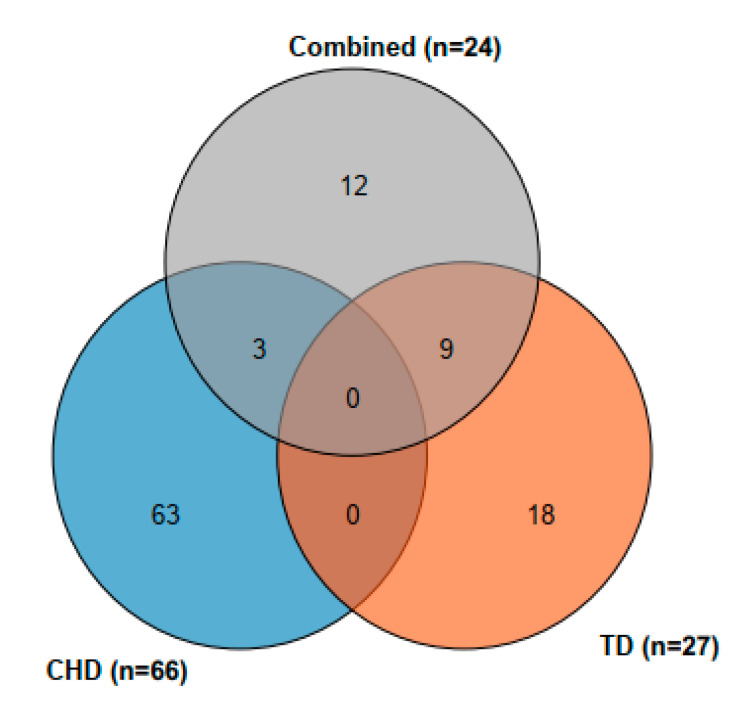
Venn diagram of SNPs detected across the three conditions (blue = combined heat and drought (CHD); red = terminal drought (TD); ash = average of CHD and TD (combined)).

**Figure 6 genes-13-00349-f006:**
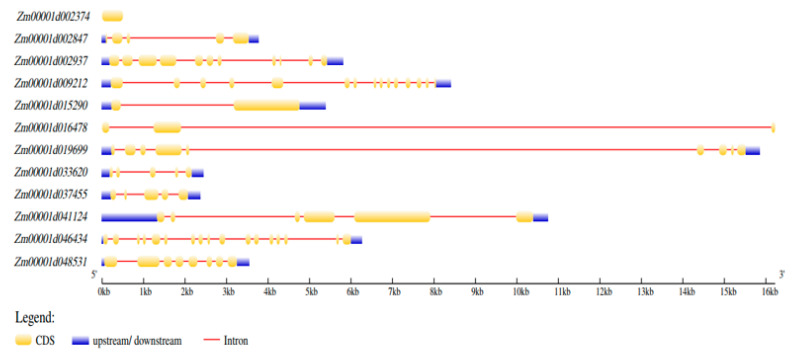
Gene structure analysis of predicted candidate genes.

**Table 1 genes-13-00349-t001:** SNPs associated with the 10 traits measured under terminal drought conditions.

Marker ^a^	Chr. ^b^	Pos. ^c^	AD	EARH	EARO	EASP	EPP	HUSKC	RL	SD	SL	GY
SNP_91146889	1	91146889	-	-	-	-	-	-	4.00 (14.58)	-	-	-
SNP_269120178	1	269120178	-	-	-	-	-	-	-	-	-	24.11 (64.79)
SNP_3121382	2	3121382	4.00 (13.77)	-	-	-	-	-	-	3.90 (13.29)	-	-
*SNP_24521844*	2	24521844	-	-	5.30 (18.09)	-	-	-	-	-	-	-
SNP_27502516	2	27502516	-	-	6.74 (23.99)	-	-	-	-	-	-	-
SNP_51109816	2	51109816	-	-	-	-	4.00 (14.71)	-	-	-	-	-
SNP_104170418	2	104170418	-	-	-	-	-	-	-	-	-	4.99 (16.77)
SNP_162101608	2	162101608	-	-	-	-	-	-	-	-	-	4.89 (16.74)
SNP_75795682	3	75795682	-	3.96 (15.12)	-	-	-	-	-	-	-	-
SNP_93411827	3	93411827	3.97 (12.84)	-	-	-	-	-	-	-	-	-
*SNP_99059711*	3	99059711	-	-	-	-	3.98 (13.45)	-	-	-	-	-
SNP_138825271	3	138825271	5.62 (18.79)	-	-	-	-	4.19 (13.84)	-	5.23 (17.32)	-	-
SNP_199184898	4	199184898	3.92 (12.53)	-	-	-	-	-	-	4.26 (13.57)	-	-
SNP_244895453	4	244895453	5.62 (18.79)	-	-	-	-	4.19 (13.85)	-	5.23 (17.32)	-	-
SNP_168561609	5	168561609	5.60 (19.79)	-	-	-	-	4.39 (14.75)	-	5.26 (18.43)	-	-
*SNP_82875264*	5	82875264	-	-	-	-	-	-	4.08 (13.86)	-	-	-
SNP_164264714	5	164264714	-	-	-	-	-	-	5.21 (18.97)	-	-	-
SNP_181875996	5	181875996	-	-	-	-	-	-	4.30 (14.52)	-	-	-
SNP_204485620	5	204485620	-	-	-	-	-	4.17 (13.22)	-	-	-	-
SNP_62970998	6	62970998	5.38 (19.96)	-	-	-	-	4.18 (15.44)	-	5.12 (18.90)	-	-
SNP_93721476	6	93721476	-	5.25 (18.10)	-	-	-	-	-	-	-	-
*SNP_125871560*	6	125871560	-	-	-	-	-	-	-	-	4.39 (14.72)	-
SNP_128658596	6	128658596	4.01 (13.60)	-	-	-	-	-	-	-	-	-
SNP_165897715	6	165897715	-	4.34 (15.42)	-	-	-	-	-	-	-	-
SNP_158184157	7	158184157	-	-	-	4.07 (13.84)	-	-	-	-	-	-
*SNP_43475091*	8	43475091	-	-	-	-	-	-	4.08 (13.86)	-	-	-
*SNP_89731392*	9	89731392	-	-	-	-	-	-	5.71 (19.26)	-	-	-

^a^ Marker names italicized are those markers detected simultaneously under terminal drought (TD) conditions, combined heat and drought (CHD) and across conditions. ^b^ Chromosome number. ^c^ Position of SNP. The value in the cell under each trait represents the peak value (phenotypic variation explained by the marker). Days to 50% anthesis (AD), ear height (EARH), ear rot (EARO), ear aspect (EASP), ears per plant (EPP), husk cover (HUSKC), root lodging (RL), days to 50% silking (SD), stalk lodging (SL), and grain yield (GY).

**Table 2 genes-13-00349-t002:** SNPs associated with the 15 traits evaluated under combined heat and drought condition plus the terminal drought condition.

Marker ^a^	Chr. ^b^	Pos. ^c^	AD	ASI	EARO	EASP	EPP	LD	PLASP	RL	SD	SL	GY
SNP_26125481	1	26125481	-	-	-	-	-	3.94 (13.84)	-	-	-	-	-
SNP_203396231	1	203396231	-	-	-	-	-	-	-	-	-	-	5.55 (19.17)
SNP_265278865	1	265278865	-	-	-	-	-	-	-	-	-	-	6.36 (22.12)
*SNP_269120178*	1	269120178	-	-	-	-	-	-	-	-	-	-	15.55 (47.77)
**SNP_11473743**	2	11473743	5.11 (17.76)	-	-	-	-	-	-	-	4.15 (14.52)	-	-
*SNP_24521844*	2	24521844	-	-	5.11 (17.42)	-	-	-	-	-	-	-	-
*SNP_27502516*	2	27502516	-	-	6.38 (22.72)	-	-	-	-	-	-	-	-
SNP_1001688	3	1001688	-	4.22 (15.56)	-	-	-	-	-	-	-	-	-
*SNP_99059711*	3	99059711	-	-	-	-	5.62 (19.08)	-	-	-	-	-	-
SNP_112491022	3	112491022	-	-	-	-	4.59 (17.42)	-	-	-	-	-	-
SNP_41272646	4	41272646	-	-	-	4.24 (13.99)	-	-	-	-	-	-	-
SNP_137415099	4	137415099	-	-	-	5.35 (17.88)	-	-	-	-	-	-	-
SNP_166608728	4	166608728	-	-	-	-	-	-	4.18 (14.24)	-	-	-	-
SNP_176971048	4	176971048	-	-	-	3.99 (13.17)	-	-	-	-	-	-	-
SNP_218713681	4	218713681	-	-	-	4.22 (14.08)	-	-	-	-	-	-	-
SNP_82875264	5	82875264	-	-	-	-	-	-	-	4.09 (13.94)	-	-	-
*SNP_164264714*	5	164264714	-	-	-	-	-	-	-	5.12 (18.43)	-	-	-
SNP_101458697	6	101458697	4.07 (13.05)	-	-	-	-	-	-	-	-	-	-
*SNP_125871560*	6	125871560	-	-	-	-	-	-	-	-	-	4.69 (15.87)	-
SNP_169223617	6	169223617	-	-	-	4.36 (15.10)	-	-	-	-	-	-	-
**SNP_51772182**	7	51772182	-	-	-	-	-	-	4.55 (16.22)	-	-	-	-
*SNP_43475091*	8	43475091	-	-	-	-	-	-	-	4.09 (13.94)	-	-	-
*SNP_89731392*	9	89731392	-	-	-	-	-	-	-	-	-	5.44 (18.44)	-
**SNP_158056460**	9	158056460	-	-	-	-	-	4.30 (15.33)	-	-	-	-	-

^a^ Marker names bolded and italicized are those markers detected simultaneously in combined heat and drought as well as the combination of this condition plus the terminal drought, and terminal drought and combined condition, respectively. ^b^ Chromosome number. ^c^ Position of SNP. The value in cell under each trait represents peak value (phenotypic variation explained by the marker). Days to 50% anthesis (AD), anthesis–silking interval (ASI), ear rot (EARO), ear aspect (EASP), ears per plant (EPP), leaf death (LD), plant aspect (PLASP), root lodging (RL), days to 50% silking (SD), stalk lodging (SL), and grain yield (GY).

**Table 3 genes-13-00349-t003:** Common SNPs detected across either CHD or TD and combined effects (means).

SNPs ^a^	Chr. ^b^	Conditions ^c^		
		CHD	TD	Combined Effects
**SNP_269120178**	1	-	YIELD	YIELD
SNP_11473743	2	AD	AD and SD	-
SNP_24521844	2	-	EARO	EARO
SNP_27502516	2	-	EARO	EARO
SNP_99059711	3	-	EPP	EPP
SNP_82875264	5	-	RL	RL
SNP_164264714	5	-	RL	RL
SNP_125871560	6	-	SL	SL
SNP_51772182	7	EASP and PLASP	PLASP	-
SNP_43475091	8	-	RL	RL
SNP_89731392	9	-	SL	RL
SNP_158056460	9	LD	LD	-

^a^ Single nucleotide polymorphisms. ^b^ Chromosomes. ^c^ CHD = combined heat and drought conditions; TD = terminal drought conditions and combined with means are averages across CHD and TD conditions. Days to 50% anthesis (AD), ear rot (EARO), ear aspect (EASP), leaf death (LD), plant aspect (PLASP), root lodging (RL), stalk lodging (SL), and grain yield (GY).

**Table 4 genes-13-00349-t004:** Candidate genes predicted in this study based on functional annotations.

SNPs ^a^	Chr. ^b^	Conditions ^c^			Gene: Position (bp) ^d^	Description ^e^
		CHD	TD	Combined Effects		
SNP_269120178	1	-	GY	GY	*Zm00001d033620*: 269012383–269014829	Peptide α-N-acetyltransferase/Protein N-terminal acetyltransferase
SNP_11473743	2	AD	-	AD & SD	*Zm00001d002374*: 11368789–11369293	Auxin induced-like protein
SNP_24521844	2	-	EARO	EARO	*Zm00001d002847*: 24476397–24480168	- (+)-neomenthol dehydrogenase/Monoterpenoid dehydrogenase
SNP_27502516	2	-	EARO	EARO	*Zm00001d002937*: 27420577–27426388	11-oxo-β-amyrin 30-oxidase/CYP72A154//Secologanin synthase
SNP_99059711	3	-	EPP	EPP	*Zm00001d041124*: 99054551–99065304	Unknown
SNP_82875264	5	-	RL	RL	*Zm00001d015290*: 82874339–82879720	Clock-associated PAS protein ZTL (ZTL)
SNP_164264714	5	-	RL	RL	*Zm00001d016478*: 164218445–164234669	Glutathione-disulfide reductase/NADPH:oxidized-glutathione oxidoreductase//Thioredoxin-disulfide reductase/Thioredoxin reductase
SNP_125871560	6	-	SL	SL	*Zm00001d037455*: 125931589–125933956	Phosphoribosylanthranilate isomerase
SNP_51772182	7	EASP & PLASP	-	PLASP	*Zm00001d019699*: 51748841–51764687	Protein-serine/threonine phosphatase/Serine/threonine specific protein phosphatase
SNP_43475091	8	-	RL	RL	*Zm00001d009212*: 43471862–43480271	Dihydrolipoyl dehydrogenase/Lipoyl dehydrogenase
SNP_89731392	9	-	RL	SL	*Zm00001d046434*: 89628666–89634933	Acyl CoA binding protein (ACBP)//Kelch motif (Kelch_1)//Galactose oxidase, central domain (Kelch_3)//Kelch motif (Kelch_5)
SNP_158056460	9	LD	-	LD	*Zm00001d048531*: 157939096–157942646	RNA helicase

^a^ Single nucleotide polymorphisms. ^b^ Chromosomes. ^c^ CHD = combined heat and drought conditions; TD = terminal drought conditions and across (combined) effects and TD conditions. Days to 50% anthesis (AD), ear rot (EARO), ear aspect (EASP), leaf death (LD), plant aspect (PLASP), root lodging (RL), stalk lodging (SL), and grain yield (GY). ^d^ Genes and their positions on each chromosome (start–end in base pair). ^e^ Annotations retrieved from https://phytozome-next.jgi.doe.gov/ (version 13) (accessed on 14 December 2021).

## Data Availability

The DArTseq datasets used in the present study have been deposited at the IITA CKAN repository.

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
