# Peer review of "Genome-Wide Association Study Reveals Genetic Architecture and Candidate Genes for Yield and Related Traits under Terminal Drought, Combined Heat and Drought in Tropical Maize Germplasm"

_genes, 2022, doi:10.3390/genes13020349_

Round 1

Reviewer 1 Report

In this paper, the association of maize SNPs and their biological features under two conditions of CHD and TD was marked through genome-wide association analysis, by using mixed linear model, and predicted the corresponding genes of the studied traits.

  1. The introduction should preferably be able to briefly describe both CHD and TD environmental conditions.
  2. This paper focuses on the maize traits associated with SNPs. In the introduction, these biological traits should be specifically expanded, and tables or diagrams should be drawn to describe the key research traits and descriptions.
  3. In the Materials & Methods section, an explanatory diagram or flowchart of the overall workflow should be added.
  4. In Results section, subsection 3.2, the colour of Figure 1 is unclear, so that the readability needs to be improved. Moreover, using a single histogram is not conducive to visually displaying the relationship between the two environmental conditions and their mean and total values.
  5. In the Results section, subsection 3.6, the data predicted by in-silico methods based on bioinformatics should be compared with the data verified by relevant experiments, and a basic accuracy statement should be made.
  6. If conditions permit, a simple online website or database can be developed to store the SNPs data associated with environmental conditions and their traits in the article, as well as the gene prediction results, so that relevant researchers can view and retrieve them easily. (optional suggestion)

Author Response

##REVIEWER 1

General comment

In this paper, the association of maize SNPs and their biological features under two conditions of CHD and TD was marked through genome-wide association analysis, by using mixed linear model, and predicted the corresponding genes of the studied traits.

#Issue 1: The introduction should preferably be able to briefly describe both CHD and TD environmental conditions.

#Response: We have devoted one full paragraph to elaborating CHD and TD in the revised manuscript (See Line 79-91). Thank you.

#Issue 2: This paper focuses on the maize traits associated with SNPs. In the introduction, these biological traits should be specifically expanded, and tables or diagrams should be drawn to describe the key research traits and descriptions.

#Response: Thank you for your suggestion. We have explained the biological traits measured for this study in the revised manuscript as you suggested (Line 150-179 and for detail  description see Supplementary Table S1).

#Issue 3: In the Materials & Methods section, an explanatory diagram or flowchart of the overall workflow should be added.

#Response: We have added flowchart as suggested into the revised manuscript (See Figure 1) in the revised manuscript.  Changes in other Figure numbers in the revised manuscript have been modified accordingly. Thank you.

#Issue 4: In Results section, subsection 3.2, the colour of Figure 1 is unclear, so that the readability needs to be improved. Moreover, using a single histogram is not conducive to visually displaying the relationship between the two environmental conditions and their mean and total values.

#Response: We have replaced Figure 1 now Figure 2 in the revised manuscript with a higher resolution one to enhance readability. This Figure gives the number of SNP markers detected on each of the ten chromosomes as well as different conditions (either CHD, TD or combined effect [average of CHD &TD]) not the relationship between the two environmental conditions and their means and total values, therefore we are of humble opinion that histogram is more appropriate. Thank you.

#Issue 5: In the Results section, subsection 3.6, the data predicted by in-silico methods based on bioinformatics should be compared with the data verified by relevant experiments, and a basic accuracy statement should be made.

#Response: Thank you for your valuable suggestion. Our study lays foundation for functional validation, therefore we are not in position to claim the actual roles of the predicted candidate genes.

#Issue 6: If conditions permit, a simple online website or database can be developed to store the SNPs data associated with environmental conditions and their traits in the article, as well as the gene prediction results, so that relevant researchers can view and retrieve them easily. (optional suggestion)

#Response: We are grateful for your suggestion, we will consider upon discussion with our team and donors.  

Reviewer 2 Report

This manuscript was conducted to identify SNPs (single nucleotide polymorphisms) that are linked to the several agronomic traits under either CHD or TD conditions or both conditions for marker-assisted breeding. The topic is interesting and the data analysis looks good. The paper could be accepted after modification of several tiny issues. Please see the comments below:

P1, L18. It should be “To breed for tolerant maize cultivars to these stresses, 162 tropical maize inbred lines were evaluated under combined heat and drought (CHD) and terminal drought (TD) conditions.” It seems this sentence has a sub-clause.

P1, L21. In the Abstract, the representative of SNP is unclear while it appeared at the first time. Please better explain the abbreviations. Although reader could get the information from the main text, it should be explained in the abstract.

P15, L375. The statement “Flowering in maize (denoted as AD and SD in this study) is an important trait in breeding for drought/heat tolerance” should be related the results found in current study in detail. What is the implication of the current study for this statement?

P15, L380-393. This paragraph has no references to support the statements or compare your results to others. Meanwhile, many statements in this paragraph had been shown in Results part, and I suggest authors should give the implication and discussions of those results, not the results itself.

Discussion section: Most of the paragraphs have no (or only a few of) literature papers to support or compare your results. I believe this section could be potentially improved in the paper.

Figure 2. The Y-label of d, f is indistinct. The resolution of these two labels is low. Figure 3 has the same problem.

Author Response

##REVIEWER 2

A brief summary

This manuscript was conducted to identify SNPs (single nucleotide polymorphisms) that are linked to the several agronomic traits under either CHD or TD conditions or both conditions for marker-assisted breeding. The topic is interesting and the data analysis looks good. The paper could be accepted after modification of several tiny issues. Please see the comments below:

#Response: We appreciate your compliment and we are motivated. Thank you. We have revised according to suggestions by your good self and another reviewer. manuscript which was omitted by error in our previous submission. Thank you.

#Issue 1: P1, L18. It should be “To breed for tolerant maize cultivars to these stresses, 162 tropical maize inbred lines were evaluated under combined heat and drought (CHD) and terminal drought (TD) conditions.” It seems this sentence has a sub-clause.

#Response: We have modified as suggested (See Line 18-20) in the revised manuscript.  

#Issue 2: P1, L21. In the Abstract, the representative of SNP is unclear while it appeared at the first time. Please better explain the abbreviations. Although reader could get the information from the main text, it should be explained in the abstract.

#Response: Thank you for your suggestion. However, the number of words allowed in abstract is limited to maximum of 200 words, hence our inability to explain fully in the abstract names of the associated markers.

Issue 3: P15, L375. The statement “Flowering in maize (denoted as AD and SD in this study) is an important trait in breeding for drought/heat tolerance” should be related the results found in current study in detail. What is the implication of the current study for this statement?

#Response: We have included the relevance of these two traits in selecting for drought and combined heat and drought tolerant materials and their implications to this current study in the revised manuscript. Please see Line 492-496. Thank you.

#Issue 4: P15, L380-393. This paragraph has no references to support the statements or compare your results to others. Meanwhile, many statements in this paragraph had been shown in Results part, and I suggest authors should give the implication and discussions of those results, not the results itself. 

#Response: Thank you for your valuable suggestion. We have added the implication of this section in the revise manuscript (See Line 504-505). As mentioned on Line 495-496 “To the best of our knowledge no GWAS has been reported for grain yield and other secondary traits in terminal drought”, making it difficult  to relate our results to existing literature.

#Issue 5: Discussion section: Most of the paragraphs have no (or only a few of) literature papers to support or compare your results. I believe this section could be potentially improved in the paper.

#Response: We have added a number of references in text of the discussion section. For example, see Line 496 and 505. Thank you.

#Issue 6: Figure 2. The Y-label of d, f is indistinct. The resolution of these two labels is low. Figure 3 has the same problem.

#Response: We have replaced Figures 2 and 3 with better resolution versions in the revised manuscript. Thank you.
